# Compression of Ultra-High Brightness Beams for a Compact X-ray Free-Electron Laser

**River Robles ***  **and James Rosenzweig**

Department of Physics and Astronomy, University of California, Los Angeles, CA 90095, USA;
rosen@physics.ucla.edu
***** Correspondence: riverrobles@ucla.edu

**Abstract:** The creation of the first X-ray free-electron laser at SLAC in 2009 introduced the scientific community to coherent photons of unprecedented high brightness. These photons were produced, however, at the cost of billion-dollar-class price tags and kilometer-scale machine footprints. This has meant that getting access to these photons is very difficult, and those who do get access do so with a strict time budget. Now, the development of critical enabling technologies, in particular high-field cryogenically cooled accelerating cavities and short-period, high-field undulator magnets, opens the door to an X-ray free-electron laser less than 30 m in length. We present here critical potential design elements for such a soft X-ray free-electron laser. To this end, simulation results are presented focusing on the problems associated with the process of bunch compression and novel ways in which those problems can be resolved.

**Keywords:** free-electron laser; bunch compression; coherent synchrotron radiation

## 1. Introduction

The first operation of the Linac Coherent Light Source in 2009 inaugurated the X-ray free-electron laser (XFEL) as an invaluable tool for scientific discovery [1]. It immediately increased the available brightness of X-ray photons by more than 9 orders of magnitude [2]. The utility of this high-quality photon source has been limited by the restricted access provided to the scientific community by the handful of existent XFELs. This small number of instruments is driven by the XFEL's large price tag and footprint. The concomitant limited access translates to restrictions on the science that can be done with these extremely high-brightness X-ray photons. Three distinct components of the XFEL can be identified that illuminate why it has traditionally been such an expensive, large-scale machine. These are the actual accelerating structure, the undulator, and the bunch compressors. At the Linac Coherent Light Source (LCLS), these devices have lengths of roughly 900 m, 120 m, and 30 m, respectively—the XFEL is a km-scale machine with a billion-dollar-class price tag [3].

This situation has produced a drive to miniaturize both the footprint and cost of the XFEL, while maintaining its core capabilities [4–6]. A solution to this challenge has recently come to maturity in recent work by the University of California, Los Angeles (UCLA) and its collaborators [7]. In this paper, we explore some critical aspects of the ultra-compact XFEL, which is expected to produce coherent X-ray photons at the level of a few percent of the LCLS in a fraction of the length, only a few tens of meters. There are a few technological advancements that make this possible. Recent work on cryogenically cooled copper accelerating cavities has shown that accelerating gradients in excess of 100 MV/m can be obtained before the onset of substantial radiofrequency (RF) breakdown [8,9]. Application of this method to a high-field photoinjector has resulted in C-band photoinjector designs providing beams with only 55 nm normalized emittance [10,11]. This technology also means that the 900 m of LCLS linac can be immediately cut down to 90 m. This can be cut down to only 9 m if one

gives up an order of magnitude in final beam energy, which can be done as long as the undulator magnet period is simultaneously shrunk. This will also immediately permit one to scale the undulator section length commensurately. Work done on high-field, short-period undulator magnets shows that this can in fact be done. Work done at UCLA has demonstrated the feasibility of mm and sub-mm-scale period undulator magnets capable of maintaining *T*-class field strengths and larger. These advanced undulators take advantage of cryogenic technology [12] and microelectromechanical system (MEMS) fabrication techniques, respectively [13].

A key problem that afflicts next generation, extremely high-brightness beam systems applied to XFELs is the bunch compressor system [10]. The bunch compressors are one of the most delicate parts of the XFEL because if they are implemented non-optimally, they can quickly lead to emitance growth and degradation of the electron beam brightness needed for lasing [14,15]. The LCLS operates with a beam of 0.4 mm-mrad normalized emittance and dedicates 30 m to compressing the beam from roughly 830 μm rms length down to 22 μm [3]. This is done while minimizing emittance growth from coherent synchrotron radiation (CSR) and uses a laser heater to minimize energy spread increases from the microbunching instability [3,16]. If one envisions using a beam with normalized emittance of only 55 nm, enabled by the above-mentioned high-field cryogenically cooled cavities, much more stringent requirements are placed on the bunch compressors.

In this work, we present progress towards start-to-end simulations of a compact X-ray free-electron laser and, in particular, discuss the difficulties associated with bunch compression for such a compact system. In Section 2, we discuss nominal design parameters for a soft X-ray free-electron laser with a total length of roughly 30 m and a two-stage bunch compression system. In Section 3, the first compression stage is presented with simulation showing successful compression by a factor of 20 while maintaining a very small projected emittance. In Section 4, the second stage is presented, which makes use of the enhanced self-amplified spontaneous emission (ESASE) scheme in order to achieve an additional order of magnitude in the beam current. Finally, in Section 5 some remaining problems are discussed.

## 2. A Design for a Compact X-ray Free-Electron Laser

### 2.1. Electron Source

The TOPGUN design, based on high-field cryogenically cooled accelerating cavities, produces a 100 pC beam with a roughly 20 A uniform current distribution and only 55 nm projected normalized emittance [10,11]. In Figure 1, the current profile, slice normalized emittance, and longitudinal phase space at the exit of the injector are shown.

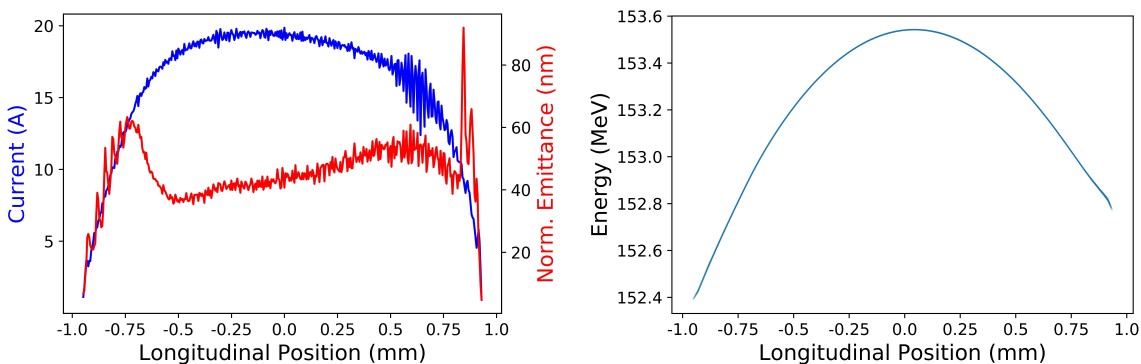

**Figure 1.** Input beam phase space from the TOPGUN design. On the left, the current profile is plotted in blue along with, in red, the slice normalized emittance. On the right, the longitudinal phase space is shown. In both plots, the beam head is to the right.

## 2.2. XFEL Parameters and Performance

In Table 1, the nominal parameters for the electron beam, undulator, and radiation are summarized for a proposed soft X-ray XFEL design. Furthermore, in Figure 2 we employ the Ming Xie fitting equations [17], modified to include the effects of space-charge by Marcus, et al. [18] to arrive at an estimate of the 3D free-electron laser (FEL) gain length as a function of beam spot size.

**Table 1.** The nominal design parameters for a soft X-ray free-electron laser (XFEL) are given. Beam parameters are based on a cryogenic-rf photoinjector, and undulator parameters are based on cryogenic, small-gap undulator work performed at the University of California, Los Angeles (UCLA) [12].

| Beam Parameter | Value | Undulator Parameter | Value |
|---|---|---|---|
| Energy U (GeV) | 1.2 | Resonant Wavelength $\lambda_r$ (nm) | 1.2 |
| Peak Current $I_{pk}$ (A) | 4000 | Undulator Period $\lambda_u$ (mm) | 9 |
| Avg. Current $I_{avg}$ (A) | 400 | Undulator Strength K | 0.97 |
| Bunch Charge Q (pC) | 100 | Pierce Parameter $\rho$ | $4.85 \times 10^{-3}$ |
| Norm. Emittance $\epsilon_n$ (μm) | 0.055 | 1D Gain Length $L_{1d}$ (cm) | 8.5 |
| Energy Spread $\sigma_\delta$ | $<10^{-3}$ | Ming Xie 3D Gain Length $L_{3d}$ (cm) | 11.5 |
| Spot Size $\sigma_r$ (μm) | 4 | Ming Xie Sat. Length $L_{sat}$ (m) | 2.3 |

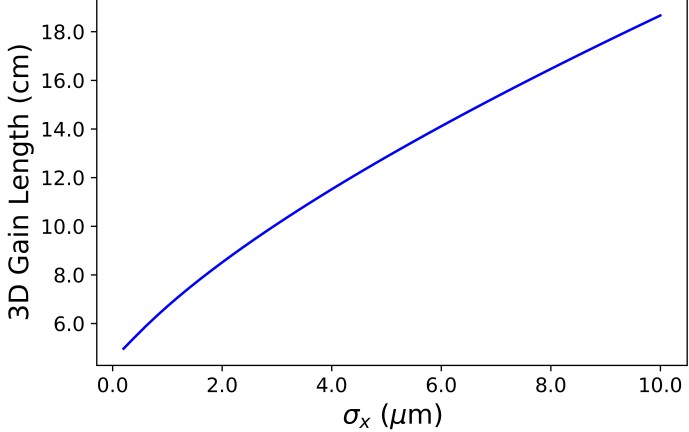

**Figure 2.** Using the Ming Xie fitting equations modified to include space-charge effects, the 3D gain length as a function of the spot size in the undulator is plotted.

We take as our working point a 4-μm beam size maintained via a strong focusing scheme based on high-gradient MEMS electromagnets [13]. At this spot size, the 3D gain length is just 11.5 cm. The saturation length can be estimated as about 20 gain lengths, which for this case gives a saturation length of 2.3 m. It is notable that this roughly 12 cm gain length holds with relative energy spread as high as $1 \times 10^{-3}$. This is a result of the fact that the Pierce parameter is large relative to standard facilities at $4.85 \times 10^{-3}$, allowing for a larger energy spread to be imposed on the beam before FEL performance suffers. Of course, this model overestimates FEL efficiency for a number of reasons; in particular, it assumes a uniform beam current of 4 kA and ignores effects such as undulator resistive wall wakefields [19,20], but it does give motivation that such an FEL has the potential to be extremely compact.

## 2.3. Summary of Beamline Layout

In the remaining sections, we attempt to bridge the gap between the electron source and the parameters listed in Table 1. In the discussions that follow, any reference to the beam emittance refers to the normalized emittance. Furthermore, color bars on particle distribution plots are intended to

gauge the relative density of particles within the plot. All simulation results downstream of the injector are found using the elegant code [21]. In the hopes of minimizing the deleterious effects of CSR, space-charge, and beamline errors, we pursue a two-stage compression from 20 A to 4 kA. The first is implemented at 400 MeV after accelerating off-crest in order to introduce a linear energy chirp on the beam. The first stage brings the beam current from 20 A up to 400 A. The second stage is not to be performed until the final beam energy of 1.2 GeV and makes use of the enhanced self-amplified spontaneous emission technique to achieve an order of magnitude increase in the local beam current from 400 A to 4 kA [22]. In the accelerating sections in between, it is assumed that we can make use of C-band cryogenically cooled cavities to achieve gradients around 125 MV/m. Using these assumed parameters, we can arrive at an estimate of the length of such a structure. The relative scales of the various elements are depicted in Figure 3. The TOPGUN injector is roughly 4 m long and accelerates the beam up to 150 MeV. Assuming nearly on-crest acceleration at 125 MV/m to 1.2 GeV, roughly 10 m of linac is required. Figure 2 implies that saturation can be expected in less than 5 m. Finally, if we ask that only 8 m is dedicated to compression, we arrive at a total length of less than 30 m. This is the goal we pursue in this analysis.

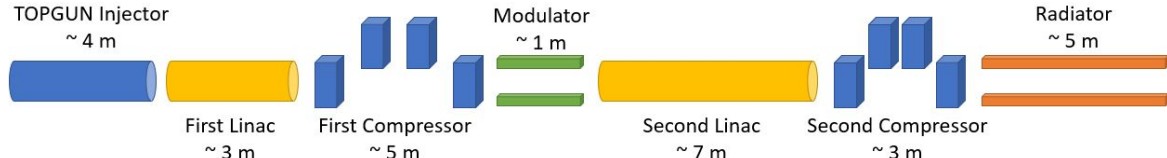

**Figure 3.** The schematic shows, in a simplified manner, the various elements present in the beamline as well as their approximate sizes. Altogether, the system should remain extremely compact at a total length of less than 30 m.

## 3. First Compression Stage

### 3.1. Design and Longitudinal Dynamics

From the gun, the beam is accelerated to 416 MeV at a phase of 76.1 degrees, where a 90-degree phase represents on-crest acceleration. At this point, a higher-harmonic cavity is used to linearize the longitudinal phase space in anticipation of the large compression by a factor of 20 to follow [23]. Because our design employs C-band RF acceleration, a traditional 11.424 GHz X-band cavity would require over 100 MeV of deceleration for linearization, which would be extremely inefficient. Instead, we design the linearizer to operate at 34.272 GHz based on development currently occurring at the Italian National Institute for Nuclear Physics (INFN) with the support of the European Union Compact Light Initiative [24]. In contrast to an X-band cavity, operation at this frequency requires only a 12 MeV voltage drop, and when operated at a phase of $-100.5$ degrees, it can prepare the beam to be compressed to a roughly linearly correlated longitudinal phase space after transport through the chicane. Figure 4 shows the longitudinal phase space and current after the action of a chicane with $R_{56} = 62$ mm. The horns on the current profile demonstrate the trade-off accepted when the 6th harmonic linearizer is used, as the beam is long enough relative to the linearizer wave that the edges are not as well-linearized as the center of the beam.

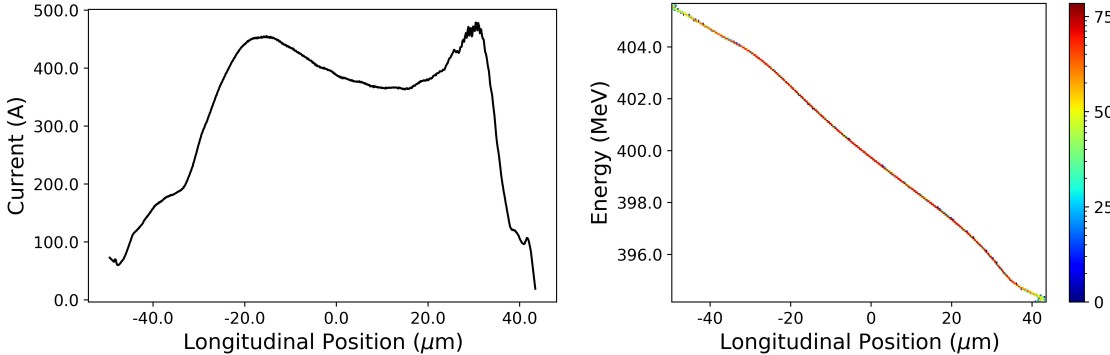

**Figure 4.** On the left, the current profile after the first compressor is shown. On the right, the longitudinal phase space is plotted. In both plots the beam head is to the right.

### 3.2. Emittance Growth in Single Chicane Design

The simplest design for the first compressor is a common four-dipole chicane; we adopt this approach first. Assuming 20-cm magnets and a final $R_{56}$ of 62 mm, the Twiss parameters $\beta$ and $\alpha$ at the entrance of the first magnet as well as the drift space between the two dipole pairs were varied in order to mininmize emittance growth arising from coherent synchrotron radiation. The optimum that was found is far from sufficient, however, because the projected emittance grows from 55 nm to 112 nm. To demonstrate the projected emittance growth, plots of the particle $x'$ coordinates are plotted against their longitudinal coordinate in the bunch in Figure 5 for cases including and excluding CSR. In the right plot, including CSR, there are clear offsets in the slice centroids leading to growth of the projected emittance. Even when the beam is collimated to accept only the particles in the full-width at half-maximum (FWHM) of the current distribution, the projected emittance still grows to nearly 80 nm.

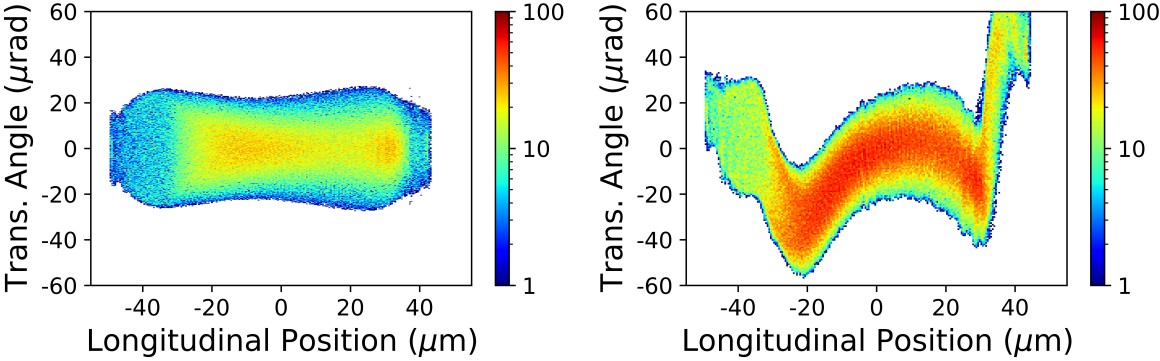

**Figure 5.** Dependence of the transverse angle on the longitudinal coordinate are shown, on the left without coherent synchrotron radiatio (CSR) effects and with CSR effects on the right. In both plots, the beam head is to the right.

### 3.3. Partial Cancellation of Emittance Growth with Double Chicane

In order to avoid the emittance growth detailed in the prior section, an alternate design was explored using an S-shaped chicane that uses two full chicanes, with the second partially canceling the CSR kicks induced in the first [25]. Because the bunch is shorter in the second chicane than in the first, typically, the second chicane is designed to have a smaller contribution to the total compression. In order to achieve this CSR-kick-canceling effect, the Twiss alpha and beta values are varied at the entrance of each chicane, as well as the bend angle and drift spacing in the first chicane, while enforcing that the total structure length remains 5.5 m and the total compression is unchanged. The results of this

approach are much better, with the projected emittance kept down to 64 nm and only 60 nm when the beam is collimated to accept only particles in the FWHM of the current distribution. Similar plots of $x'$ versus longitudinal position are shown in Figure 6, and the optimum parameter set is listed in Table 2. The optimum twiss parameters can be arranged with quadrupoles between the linacs upstream of the compressor and a quadrupole triplet between the two chicanes. The slices are now seen to be well-aligned even after the CSR kicks from the chicanes, and the emittance is well-preserved.

**Table 2.** The optimal S-chicane parameters for emittance growth cancellation are reported, resulting in the compression by a factor of 20 needed in the first stage. $\beta_x$ and $\alpha_x$ refer to the beam Twiss parameters, and the drift length refers to the drifts between the outer bend magnet pairs.

| First Chicane | Value | Second Chicane | Value |
|---|---|---|---|
| Bend Angle $\theta_1$ (deg) | 8.3 | Bend Angle $\theta_2$ (deg) | 4.2 |
| Magnet Length $L_{b1}$ (m) | 0.2 | Magnet Length $L_{b2}$ (m) | 0.2 |
| Drift Length $L_{d1}$ (m) | 1.24 | Drift Length $L_{d2}$ (m) | 0.26 |
| $R_{56}$ (mm) | 57.7 | $R_{56}$ (mm) | 4.3 |
| Entrance $\beta_x$ (m) | 11.5 | Entrance $\beta_x$ (m) | 5.0 |
| Entrance $\alpha_x$ | 3.7 | Entrance $\alpha_x$ | 2.5 |

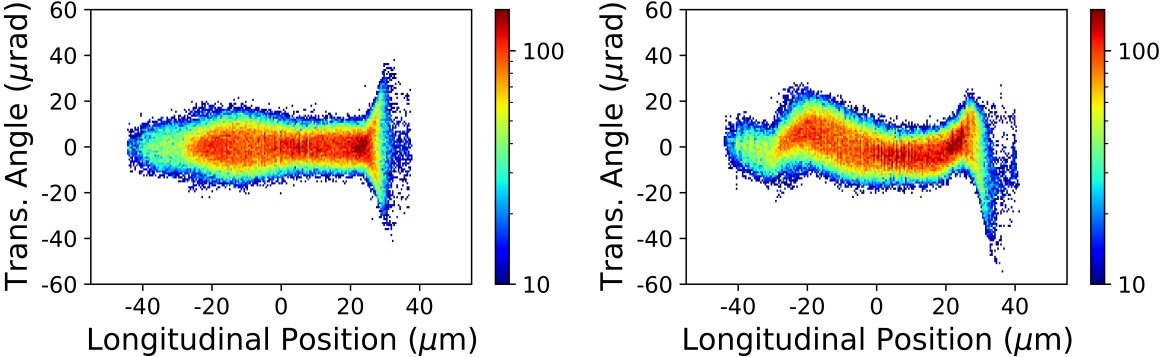

**Figure 6.** The variation of transverse angle with longitudinal bunch coordinate are shown for the S-shaped chicane design for the first compressor, on the left without CSR effects and with CSR effects on the right. In both plots, the beam head is to the right.

## 4. Second Compression Stage

### 4.1. Review and Modification of the ESASE Technique

The method of enhanced SASE was proposed in 2005 by Zholents in order to reduce FEL gain length by increasing peak current [22] and was recently proposed for the preservation of the emittance in ultra-high-brightness beams obtained from cryogenic high-field photoinjectors [10]. The beam is co-propagated with an IR laser inside of a short undulator magnet with just a few periods. If the IR laser wavelength is chosen to interact resonantly with the beam inside of the undulator, then it can impart a sinusoidal energy modulation on the beam that can be converted to density modulation inside of a small chicane. This scheme has recently been explored in proof-of-principle experiments at the LCLS [26]. In addition to the proposal for use of ESASE in advanced photoinjectors, during studies of a 0.3 angstrom wavelength XFEL at Los Alamos National Laboratory (LANL), it was realized that this same technique could be used to bring an XFEL to its baseline current while avoiding a large second stage compressor. This second stage may introduce problems like microbunching instability and further CSR effects [27]. Additionally, increasing the peak current in this way does nothing to the average current in the beam, which has been shown to suppress gain degradation from undulator resistive wall wakes in previous ESASE studies [28]. This solution is ideal for the compact XFEL because of its ability to shrink the size of the second bunch compressor and because the resistive wall

wakes are exacerbated by the narrow gap in the short-period undulator [19,20]. Because it departs notably from the goals of the traditional ESASE method, the LANL approach is termed laser-assisted bunch compression (LABC) [29]. Where an ESASE design pushes the peak current as high as possible, an LABC design will opt to overcompress the microbunches provided by the laser in order to achieve a moderate peak current while maximizing the length and flatness of the current profile inside of the microbunches. This avoids energy reduction from slippage effects that can become significant in short microbunches.

### 4.2. Application of the LABC Method to the Compact XFEL

The longitudinal phase space after dechirping the beam after the first bunch compressor is shown in Figure 7. The rms energy spread of the bunch is 23 keV. We choose to modulate the beam energy at 400 MeV right after the first bunch compressor because at higher energy it becomes more difficult to match the resonance condition to allow for the modulation to occur. We consider here a 10-µm laser wavelength in order to mitigate gain degradation from slippage in the undulator.

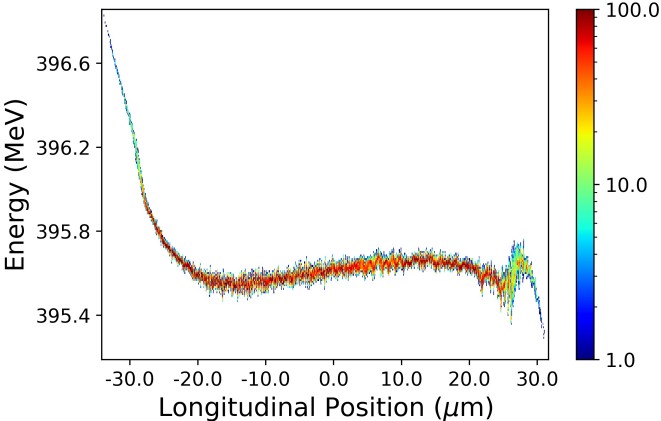

**Figure 7.** The longitudinal phase space after the first bunch compressor is shown after the remaining energy chirp was removed using a passive dechirper cavity. The beam head is to the right.

The laser modulation can be modeled as a transformation of the form $\gamma_f = \gamma_i + A\sigma_\gamma \sin(k_l z_i)$, where $\sigma_\gamma$ is the rms energy spread of the beam, $k_l$ is the laser wavenumber, and $A$ is the modulation amplitude in units of the beam's uncorrelated energy spread. The chicane that follows then gives a transformation of the form $z_f = z_i + B(\gamma_i - \bar{\gamma})/(\sigma_\gamma k_l)$, where $\gamma_i$ is the Lorentz factor of the modulated particles, $\bar{\gamma}$ is the mean Lorentz factor of the beam, and $B = R_{56}\sigma_\gamma k_l/\bar{\gamma}$ is a dimensionless parameter defined so that peak compression of the microbunches occurs at $B = 1/A$ [30]. In Figure 8, the peak current of the microbunches is plotted as a function of $A$ and $B$.

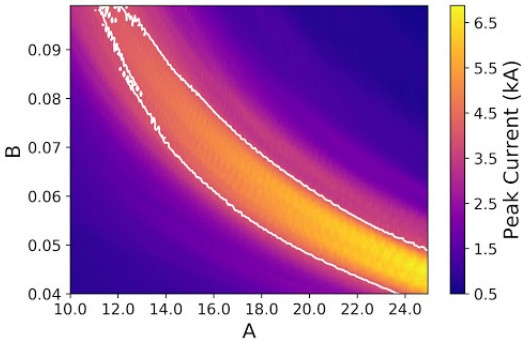

**Figure 8.** Peak current of the microbunches as a function of normalized modulation amplitude and chicane compression. The white contour line denotes 4 kA current.

In Figure 8, the white contour line marks 4 kA peak current. The lower contour represents the undercompression case, where the upper contour represents overcompression. In general, the overcompressed microbunches are, of course, longer. After scanning over this contour, the flattest microbunch is found to occur where $A = 19.13$ and $B = 0.0635$. These optimal dimensionless parameters are realized using a laser modulator and chicane whose details are listed in Table 3. The longitudinal phase space and current profile of one microbunch of this optimal working point are shown in Figure 9. The blue vertical lines in the current profile plot indicate the positions where the current drops to half of its peak value. There is a roughly 200-nm flat-top with approximately 4 kA current and a drop-off on either side giving a FWHM microbunch length of 450 nm. The effects of CSR are neglected in these plots.

**Table 3.** The parameters for the laser modulator and final chicane are given for the optimal dimensionless modulation amplitude and compressor strength discussed in the text.

| Laser Modulator | Value | Chicane | Value |
|---|---|---|---|
| Beam Energy U (MeV) | 400 | Beam Energy U (GeV) | 1.2 |
| Undulator Period $\lambda_u$ (cm) | 15 | Bend Angle $\theta$ (deg) | 2.76 |
| Peak Undulator Field $B_u$ (T) | 0.87 | Magnet Length $L_b$ (m) | 0.2 |
| Number of Periods $N_u$ | 5 | Drift Length $L_d$ (m) | 1 |
| Laser Wavelength $\lambda_l$ ($\mu$m) | 10 | $R_{56}$ (mm) | 5.27 |
| Laser Waist $w_l$ (mm) | 0.6 | | |
| Laser Peak Power $P_L$ (MW) | 60 | | |

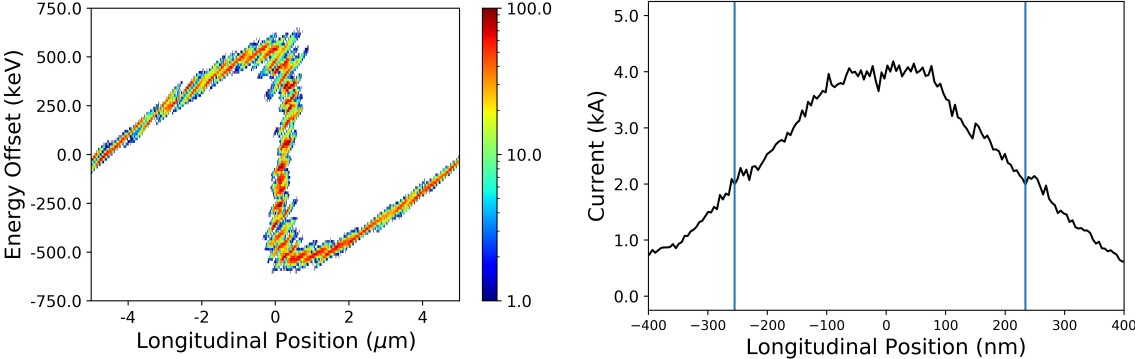

**Figure 9.** Optimal modulation amplitude and compression from the peak current contour with $A = 19.13$ and $B = 0.0635$. On the left, the longitudinal phase space of one microbunch shown is at 1.2 GeV, nominally at the entrance of the undulator. On the right, the associated current profile is plotted with the blue vertical lines indicating where the current falls to half of its peak value. In both plots, the beam head is to the right.

### 4.3. CSR Effects in the Second Bunch Compressor

As already noted, CSR effects are not yet taken into account in the second bunch compressor. It is expected that because of the smaller $R_{56}$ required for the LABC compressor, CSR effects will be substantially mitigated. Additionally, the short length of the microbunches means that the 1D models of CSR traditionally used in codes such as elegant may begin to lose fidelity, so a 3D code should be used to confirm any results obtained there. As of yet however, conclusive studies on CSR in the second bunch compressor have not been completed.

### 5. Discussion

In this paper, we have discussed a key element needed in enabling an ultra-compact X-ray free-electron laser. We have presented simulation results that suggest that the beam emittance may be able to be preserved in strong bunch compressors as long as novel compression architectures

are adopted. Although the LABC compression approach discussed here is promising, more robust simulations of 3D coherent synchrotron radiation effects are required in order to verify that the assumption that a weaker chicane leads to less harmful CSR effects is valid. This study was performed for a soft X-ray design; a nominal hard X-ray design, as discussed in Ref. [7], utilizes a yet smaller chicane necessitated by the shorter laser modulator wavelength, thereby systematically mitigating potential CSR issues. The authors are in collaboration with both LBNL and LANL in an effort to make these high-fidelity simulations a reality.

**Author Contributions:** Conceptualization, R.R. and J.R.; investigation, R.R.; resources, J.R.; writing–original draft preparation, R.R.; writing–review and editing, J.R.; visualization, R.R.; supervision, J.R.; project administration, J.R.; funding acquisition, J.R.

**Funding:** This research was funded by the US DOE Office of High Energy Physics through contract DE-SC0009914 and US NSF Award PHY-1549132, the Center for Bright Beams.

**Conflicts of Interest:** The authors declare no conflict of interest.

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
