# Peer review of "Compression of Ultra-High Brightness Beams for a Compact X-ray Free-Electron Laser"

_instruments, doi:10.3390/instruments3040053_

Round 1

Reviewer 1 Report

cv

This paper proposes compression scheme of ultra-high brightness beams for a compact X-ray FEL. The main parameters of the compact X-ray FEL are summarized in Table 1. The 100 pC beam with peak current of 20 A and normalized emittance of 55 nm at energy of 150 MeV, which can be produced from TOPGUN, is 200 times compressed in a two-stage bunch compression without largely deteriorating the emittance.
The first stage increases the beam current from 20 A to 400A at 400 MeV. The second stage increases the beam current from 400A to 4000A with ESASE method.

The presented simulation results seem valid. I will recommend publication of the manuscript after some revisions.

In page 3, authors take 4 m beam size in an undulator. This seems to require beta function of 0.7 m. Is this possible over an entire undulator of 2.3 m long ?

Please use the same x and y scales for Figs. 4 and 5 to see the difference clearly.

Please provide the design parameters of the first bunch compressor such as bending angles of the S-shaped magnets, magnet lengths, and drift lengths between magnets.
Please describe the status of development of the cavity and RF power supply for the linearlizer operated at 34.272 GHz.

Please elaborate on the proposed ESASE scheme. No undulator parameter for energy modulation is given. Please provide the laser parameters necessary for the ESASE together with references supporting those parameters.

Please provide a figure of the proposed two-stage bunch compressor to show the total size.

Reviewer 2 Report

Some comments and recommendations for the manuscript instruments-601715

September 2019

This article reports on compactification of existent large-scale x-ray FELs (km-scale), in which the length may be less than 30 m. In terms of the effective compactification, three key technologies may be mainly required, that is, high-field cryogenically-cooled accelerating cavities, short-period high-field undulators, and dedicated bunch compression systems.

The bunch compression system is one of the key technologies to maintain high-brightness electron beams without any degradation during the bunch compression due to CSR. In this article, the authors particularly focus conceptual design on the effective bunch compression system with two dedicated compression stages based on simulations (although not full 3D). The results show that any emittance growth due to CSR can be clearly minimized. Thus, their results will open the door to realize compact FELs in soft x-ray region.

I believe that this paper gives important steps and wide applications to not only academic areas but also industrial and medical areas. This article is worthwhile in publication in Instruments.

Thus, as a reader of the interesting article, some minor amendments are suggested to be revised in the following, although no major revisions are required.

Minor revisions required

In the 33th line of Introduction, it should be inserted between numerical value and unit, that is, “…. can be immediately cut down to 90 m”. In the 34th line of Introduction, it should be inserted between numerical value and unit, that is, “…. can be cut down to only 9 m….”. In Fig.1 of section 2.1, for both on the left and right plots, it should be specified which direction in longitudinal position corresponds to the beam head. It should be clarified in the figure caption. In the 74th line of section 2.2, it may be “It is notable that….”. It should be checked. In references it seems that there are two different kinds of journals, while these two journals are the same, that is, “Physical Review Accelerators and Beams” and “Physical Review Special Topics - Accelerators and Beams”. It should be unified to the latter. In Fig.3 of section 3.1, for both on the left and right plots, it should be specified which direction in longitudinal position corresponds to the beam head. It should be clarified in the figure caption. In the 103th line of section 3.1, the institute name, INFN, should be firstly written in full, that is, …. (INFN). In Fig.3 of section 3.1, it should be specified what the colorbar means on the right figure. In the 114th line of section 3.2, it should be inserted between numerical value and unit, that is, “…. can be immediately cut down to 62 mm”. In Fig.4 of section 3.2, it should be specified what the colorbars mean on both the left and right figures. In Fig.5 of section 3.2, it should be specified what the colorbars mean on both the left and right figures. In Fig.6 of section 4.1, it should be specified what the colorbar means on the figure. And it should be specified which direction in longitudinal position corresponds to the beam head. In the 164th line of section 4.2, it should be inserted between numerical value and unit, that is, “We consider here a 10 μm…. ”. In Fig.8 of section 3.2, it should be specified what the colorbar means on the left figure and it should be also specified what the two solid lines mean on the right figure. In the 183th line of section 4.3, it may be “… the short length of the microbunches means…”. In the 189th line in Discussion, there are unnecessary spaces before the period of this sentence.

Author Response

Thank you for a very thorough review. All of the changes regarding grammatical/spelling errors have been made as requested. Additionally, all "Physical Review Accelerators and Beams" references have been changed as suggested. All captions involving plots of particle distributions have been updated to note that the beam head is to the right in the plot. In line 83 a sentence has been added to clarify the meaning of the color bars. 

Reviewer 3 Report

The manuscript presents an overview of the authors' work aimed at reducing the size and cost of an x-ray FEL facility, with the original results being focused on the bunch compression scheme.  I believe that the paper will be more or less suitable for publication, once a few details are more fully explained.  I list these below:

1) Table 1 has a note above it that reads "You need to make a much more complete parameter list." I agree with this assessment, and once additional parameters including, e.g., energy spread, undulator length, etc. are included the note should be deleted.

2) The authors appear to consistently use "emittance" when referring to the normalized emittance.  I'm not sure that they need to add "normalized" everywhere, but I do think they should be a bit more careful here.

3) I do not believe that it is only the average current that controls beam degradation due to resistive wall wakes as implied in lines 150-151. The peak current and its profile will also play a role.  The authors should either provide more evidence here or soften their statement.

4) The laser and chicane parameters are only given in dimensionless form.  Please supply the associated "real world" quantities, in particular the requires laser power or intensity and R56.

5) The authors hint at CSR calculations for the 2nd bunch compressor and the associated issues, but it is not unclear whether the "results obtained here" (line 185) actually include CSR in final chicane.  Please clarify if what is being presented is the result of a preliminary 1D calculation that may need refinement or not.
